# The Pathways Linking to Sleep Habits among Children and Adolescents: A Complete Survey at Setagaya-ku, Tokyo

**DOI:** 10.3390/ijerph18126309

**Published:** 2021-06-10

**Authors:** Shingo Noi, Akiko Shikano, Ryo Tanaka, Kosuke Tanabe, Natsuko Enomoto, Tetsuhiro Kidokoro, Naoko Yamada, Mari Yoshinaga

**Affiliations:** 1Research Institute for Health and Sport Science, Nippon Sport Science University, Tokyo 158-8508, Japan; shikano.a@nittai.ac.jp (A.S.); qnzb.f.f@gmail.com (R.T.); k.tanabe@thu.ac.jp (K.T.); kidokoro@nittai.ac.jp (T.K.); 2School of Health and Sport Science, Osaka University of Health and Sport Science, Osaka 590-0496, Japan; 3Faculty of Modern Life, Teikyo Heisei University, Tokyo 164-8530, Japan; 4Doctoral Programme in Health and Sport Science, Nippon Sport Science University, Tokyo 158-8508, Japan; natsurun718@gmail.com; 5Faculty of Sport Science, Nippon Sport Science University, Tokyo 158-8508, Japan; n-yamada@nittai.ac.jp; 6Faculty of Pharmaceutical Science, Showa Pharmaceutical University, Tokyo 194-8543, Japan; yosinaga@ac.shoyaku.ac.jp

**Keywords:** school student, physical activity, screen time, parental lifestyle, neighborhood social capital

## Abstract

It has been noted that Japanese children sleep the least in the world, and this has become a major social issue. This study examined the pathways linked to sleep habits (SH) among children and adolescents. A questionnaire-based survey was conducted in March 2019 on children and their parents at all 63 public elementary and 29 public junior high schools in Setagaya-ku, Tokyo. For the analysis, 22,385 pairs of children–parent responses (valid response rate: 68.8%) with no missing data were used. This survey collected data on SH, physical activity (PA), screen time (ST) for the child, and lifestyle and neighborhood social capital (NSC) for the parents. Moreover, the pathways linking ‘NSC’ → ‘parental lifestyle’ → ‘child’s PA/ST’ →‘child’s SH’ were examined through structural equation modeling. The results indicated that children’s SH were affected by their PA and ST and influenced by the lifestyle of their parents and the NSC that surrounds them. Thus, we concluded that it is necessary to provide direct interventions and take additional measures with regard to parent lifestyle and their NSC to solve persistent sleep problems in children.

## 1. Introduction

The ‘era of sleep difficulty’ has arrived for humankind. The fact that the World Health Organization’s (WHO) Regional Office for the European Centre for Environment and Health held an expert meeting on sleep disorders in Bonn, Germany, in January 2004 confirms this [1]. Moreover, at the WHO General Assembly held in May 2019, ‘sleep–wake disorders’ were registered in the International Classification of Disease, 11th Revision (ICD-11) [2]. In other words, unhealthy sleep habits (SH) are a very common problem, which should be considered a universal health issue.

The shorter sleep duration among Japanese people is a serious problem, and children are no exception. According to the Japan School Health Society [3], sleep duration for elementary school students over the last three decades has shortened by approximately 10–30 min, and 40 min or more for junior high school students. Moreover, when compared to data from other countries [4], it was found that Japanese children sleep the least in the world. Not only that, in the ‘concluding observations on the combined fourth and fifth periodic reports of Japan’, presented by the United Nations Committee on the Rights of the Child (CRC), efforts to secure necessary rest and leisure time for all children were recommended [5].

In this situation, a national campaign to promote early bedtime, early wake-up, and breakfast in Japan is underway. However, various survey results show that the campaign and other efforts during this period have not succeeded in significantly improving children’s lifestyles. Thus, it is necessary to examine the reasons behind children’s sleep problems comprehensively.

According to Bronfenbrenner’s Model [6], a child’s development is influenced by multiple levels of the surrounding environment, including the individual, interpersonal, and social. At the individual level, previous studies have shown that physical activity (PA) [7,8,9,10] and screen time (ST) [11,12,13] are both associated with sleep outcomes. At the interpersonal level, evidence suggests that parental lifestyle is significantly associated with children’s PA and ST [14,15,16,17]. Additionally, growing evidence has shown that neighborhood social capital (NSC), which is defined as the resources accessed by individuals as a result of their membership of a network or a group, is associated with parental lifestyle [18,19], which in turn influences children’s SH.

These previous studies were advantageous to our research. However, while they demonstrated individual associations, none have been examined simultaneously. Structurally, the relationships among social capital and parental lifestyle, PA, and ST of children are also yet to be examined. Structural equation modeling (SEM) has become a standard tool in many scientific disciplines for investigating the plausibility of theoretical models that might explain the interrelations among a set of variables [20].

Therefore, using SEM, this study examined how the SH of Japanese children and adolescents is influenced by surrounding environmental factors including NSC, parental lifestyle, and their own lifestyle behavior (i.e., PA and ST). Based on the Japan School Health Society [3], this study also aimed to examine whether the hypothetical model was different according to sex and school stage.

## 2. Materials and Methods

### 2.1. Ethics Approval

The study’s design was approved by the ethics committee of the Nippon Sport Science University (approval No. 015-H075). Participants were informed of the purpose and contents of the survey, informed about their right to withdraw from the study at any time, and were provided with complete assurance regarding the confidentiality of their data. Returning completed questionnaires meant that participants had given their informed consent.

### 2.2. Participants and the Survey Period

The survey was conducted in Setagaya-ku, Tokyo. The focus group included parents and their children enrolled in the third grade or above at all public elementary schools (63 schools) and all public junior high schools (29 schools), as of March 2019. For the analysis, 22,385 pairs of child–parent responses (valid response rate: 68.8%) with no missing data were used.

Setagaya Ward, the target area of this study, is located at the southwestern end of Tokyo’s 23 wards (approximately 139° 39 min east longitude, 35° 38 min north latitude), 15 km from the city center. It has an area of 58.05 km^2^ (east-west, about 9 km; north-south, about 8 km). Additionally, according to the Basic Resident Register, as of June 2019, the population (915,215 people) and the number of households (485,834 households) were the highest among Tokyo’s 23 wards. The population composition ratio of this area is 11.8% for those aged 0–14 years, 68.1% for 15-to-64-year-olds, and 20.1% for those over 64 years of age.

### 2.3. Hypothetical Model

As previously mentioned, no studies have simultaneously and structurally examined the relationships among social capital and parental lifestyle, PA, and ST of children. SEM has become a standard tool in many scientific disciplines for investigating the plausibility of theoretical models that might explain the interrelations among a set of variables [20]. In this study, we developed the following hypothetical model about the relationship between NSC, parental lifestyle, children’s ST, PA, and SH.

#### 2.3.1. Neighborhood Social Capital and Parental Lifestyle

McCloskey and Pei [19] found that NSC relationship has an impact on maternal mental health and that efforts to strengthen neighborhood social cohesion may improve outcomes by reducing childcare stress. Furthermore, Chen et al. [18] showed that social relationship capital influences PA for those aged 15 to 69 years, noting that social participation expands the social network of residents and positively influences PA and nutrition. These previous studies showed that NSC was linked to parental health and lifestyle. Therefore, the hypothesis regarding NSC and parental lifestyle was as follows:

**Hypothesis** **1.**
*NSC has a direct effect on parental lifestyle.*


#### 2.3.2. Parental Lifestyle and Children’s Lifestyle

Several studies have previously reported that parents’ lifestyle habits influence those of their children. For example, Schoeppe et al. [21] found a positive relationship between parents’ sports participation and children’s PA. This study suggested that both maternal and paternal sports participation was positively associated with children’s engagement in leisure-time PA. In addition, Poulain et al. [22] noted that children’s ST was significantly related to their mothers’ ST, which was reciprocally influenced by the use of electronic media by parents and children together. Based on this, we hypothesized the following:

**Hypothesis** **2.**
*Parental lifestyle has a direct effect on children’s PA.*


**Hypothesis** **3.**
*Parental lifestyle has a direct effect on children’s ST.*


#### 2.3.3. Physical Activity/Screen Time and Sleep Habits

PA has been reported to be effective in improving sleep problems. Stone et al. [8] found that engaging in higher intensity PA throughout the week can help children fall asleep more quickly and maintain healthy SH. In addition, studies on Japanese children have pointed out the relationship between the amount of PA during the day and sleep [7,10]. Conversely, ST is one of the factors that interferes with sleep. Van den Bulck [11] reported that children who spent more time playing computer games went to bed later on weekdays and weekends, and woke up later on weekends. It has also been argued that prolonged media use, not only delays bedtime, but also reduces sleep quality, as it arouses the body psychologically and physically [12]. Therefore, we formulated the following hypotheses.

**Hypothesis** **4.**
*PA has a direct effect on children’s SH.*


**Hypothesis** **5.**
*ST has a direct effect on children’s SH.*


### 2.4. Questionnaire and Procedure

We created a self-administered questionnaire and asked the children and their parents to respond to the survey anonymously. The questionnaires were delivered and collected by the Setagaya Ward Board of Education from each school and homeroom teacher.

The questionnaire for the children focused on ‘falling asleep’, ‘nocturnal awakening’, and ‘awakening’, based on research that has pointed out that sleep problems are an important indicator of SH [23,24]. The specific questions were (1) subjective ease of initiating sleep: ‘Do you have difficulty falling asleep at night?’; (2) subjective estimate of waking up during the night: ‘Do you wake up during the night after you have gone to sleep?’; and (3) subjective ease of waking (rating the level of difficulty the participant has waking up in the morning): ‘Do you have difficulty waking up in the morning?’. Subjects were requested to answer ‘yes’ or ‘no’ to the three sleep problem questions. In addition, the questionnaire enquired about bedtime and wake-up time, and nocturnal sleep duration was also calculated from these records. Moreover, as PA habits are presumed to be related to sleep problems, engagement in play/exercise (EX) habits before class, after lunch, after school, and total EX time in a week were requested. Questions on ST habits covered electronic use, such as hours spent on video games, TV/video/DVD, mobile/smartphone, and tablet/personal computer (PC) utilization per day.

The questionnaire for parents focused on bedtime and wake-up times on weekdays, EX habits during the year (daily, 5–6 days in a week, 3–4 days in a week, 1–2 days in a week, several times in a month), and the frequency of reading books (RB) per month (0, 1, 2–3, 4–7, 8–11, or 12 books or more). Nocturnal sleep duration was calculated from bedtime and wake-up time records. In addition, five questions were asked about NSC. Among these, three items (local people are reliable, are strongly related, and are happy to help neighbors) were developed by Nawa et al. [25], with reference to Sampson et al. [26] and Jung et al. [27]. Two items (I want to participate in local events, and local people are supportive of children attending school) were developed by us considering the characteristics of Japanese culture, where almost all children walk to school, and the community organizes various events. A five-point Likert scale was used to collect the answers (yes, somewhat yes, neither, somewhat no, and no).

### 2.5. Data Analysis

In this study, the following three points were mainly analyzed based on the obtained data.

The first was the examination of the lifestyle characteristics of the subjects. In this examination, we calculated the mean ± standard deviation (SD) or the distribution of the answers to each question according to sex and grade level, then compared the findings with those from previous studies and examined the characteristics of the participating children’s lifestyles.

The second was the examination of the validity of latent variables. For this examination, confirmatory factor analysis (CFA) was used. The responses to the observation variable of ‘NSC’ were scored as ‘yes’ (5 point), ‘somewhat yes’ (4 points), ‘neither’ (3 points), ‘somewhat no’ (2 points), and ‘no’ (1 points). Similarly, in parental lifestyle, the response to the observation variable ‘parent’s bedtime’ (>21:00 = 1 point, >22:00 = 2 point, >23:00 = 3 point, >24:00 = 4 point, 24:00≦ = 5 point), the observation variable ‘parent’s wake-up time’ (>5:30 = 1 point, >6:00 = 2 point, >6:30 = 3 point, >7:00 = 4 point, 7:00≦ = 5 point), the observation variable ‘EX habits’ (every day = 5 point, 5–6 days in a week = 4 points, 3–4 days in a week = 3 points, 1–2 days in a week = 2 points, several times a month = 1 points), and the observation variable ‘RB habits’ (No book = 1 point, 1 book = 2 points, 2–3 books = 3 points, 4–7 books = 4 points, 8–11 books = 5 points, 12 books or more = 6 points) were scored. Furthermore, in children’s lifestyle, the response to the observation variables ‘before class’, ‘after lunch’, ‘after school’ (yes = 1 point, no = 0 points), the observation variable ‘EX time in a week’ (continuous variable), and the observation variables ‘video game’, ‘TV/video/DVD’, ‘smartphone’, ‘table/PC’ (continuous variable) were used. Finally, in children’s SH, the observation variables ‘falling asleep’, ‘nocturnal awakening’, and ‘awakening’ (yes = 1 point, no = 0 point) were scored. The observation variable ‘bedtime’ and ‘wake-up time’ were scored in the same categories as the above ‘parent’s bedtime’ and ‘parent’s wake-up time’.

The third was to verify the hypothetical model. In this examination, we verified the hypothetical model shown in Figure 1 through SEM, while we confirmed the robustness using the bootstrap method [28]. Here, it is undeniable that the verification results may differ depending on sex and school level. Therefore, we constructed models considering sex and school stage differences and verified them by simultaneous multiple-group SEM. In this analysis, we confirmed that the configural invariance model (CIM) was the same regardless of sex and school level, and then we also compared the CIM and equality constraints model (ECM), which imposed equality constraints on each latent variable.

The maximum likelihood method was used to estimate the parameters, and the goodness-of-fit index (GFI), adjusted goodness-of-fit index (AGFI), comparative fit index (CFI), and root mean square error of approximation (RMSEA) were used. We also used the Akaike information criterion (AIC) and chi-square (CMIN) to compare the models in the simultaneous multiple-group SEM and comprehensively determined the suitability of the models; for example, GFI and AGFI were 0.9 or above, RMSEA was 0.8 or below [25]. These analyses were conducted using IBM SPSS Amos ver. 25.0.

## 3. Results

Table 1 shows the response results for children’s SH, PA, and ST, and Table 2 shows the response results for parents’ lifestyles and NSC. After analyzing the children’s SH, it was determined that there was no difference in the average wake-up times according to sex and grade level, while bedtimes were noticeably delayed in the upper grades (3–4 grade: boys 21:47 ± 43, girls 21:52 ± 43; 5–6 grade: boys 22:11 ± 48, girls 22:19 ± 48; and 7–9 grade: boys 23:28 ± 82, girls 23:35 ± 76), and the nocturnal sleep duration was reduced accordingly. In addition, for both boys and girls, the higher the grade in school and the greater the number of people with an ST of 2 h or more, the higher the number with bad awakening habits and the lower the exercise habits before class and after lunch.

Next, the validity of each latent variable was examined by CFA (Figure 2). The estimated value of ‘nocturnal awakening’ was not significant, so it was excluded from the observation variable of the latent variable ‘SH’. As a result, as shown in Figure 2, the goodness-of-fit in each latent variable was adopted.

After the above procedure, the hypothetical model in this study was examined by SEM (Figure 3). As Figure 3 demonstrates, the goodness-of-fit values in this model were GFI = 0.958, AGFI = 0.946, CFI = 0.860, and RMSEA = 0.050, and all estimated values were significant. Incidentally, the result of confirming the robustness, including the model described in Figure 4 and Figure 5, was as shown in Appendix A.

Furthermore, the results of the examination of sex differences in this model are presented in Figure 4. As a result, the goodness-of-fit indices in CIM, which reports the same configural invariance regardless of sex, were GFI = 0.957, AGFI = 0.945, CFI = 0.859, RMSEA = 0.035, AIC = 10,636.604, and CMIN = 10,432.604, and the placement invariance of the model was confirmed. Thus, to examine the sex differences from the estimated values of the model, we applied an equality constraint to the estimated values between each latent variable. The goodness-of-fit indices were GFI = 0.956, AGFI = 0.944, CFI = 0.856, RMSEA = 0.035, AIC = 10,832.532, and CMIN = 10,638.532, and the estimated values of ‘NSC’ → ‘parental lifestyle’ (boys = 0.16, girls = 0.13), ‘parental lifestyle’ → ‘PA’ (boys = 0.27, girls = 0.14), ‘parental lifestyle’ → ‘ST’ (boys = −0.12, girls = −0.34), and ‘ST’ → ‘SH’ (boys = 0.51, girls = 0.64) indicated significant differences. The goodness-of-fit indices were better with CIM than with ECM (Table 3). Similarly, the results of the examination of school grade differences in this model are presented in Table 3 and Figure 5. As a result, the goodness-of-fit indices in CIM were GFI = 0.962, AGFI = 0.951, CFI = 0.867, RMSEA = 0.034, AIC = 16,249.771, and CMIN = 16,045.771, and the placement invariance of the model was confirmed. Thus, to examine the school grade differences for the estimated values of the model, we applied an equality constraint to the estimated values between each latent variable. In this case, the goodness-of-fit indices were GFI = 0.961, AGFI = 0.950, CFI = 0.864, RMSEA = 0.034, AIC = 16,588.953, and CMIN = 16,394.953, and the estimated values of ‘ST’ → ‘SH’ (elementary school = 0.26, junior high school = 0.56) denoted significant differences.

From the above facts, for both sex and school grade differences in this model, the goodness-of-fit indices for CIM were better than ECM (Table 3), and the estimated values shown in Figure 4 and Figure 5 were all significant.

## 4. Discussion

The nocturnal sleep duration of children in the present study (Table 1) was extremely short compared to that of children from other countries [4,29]. Additionally, all sleep durations shown in Table 1 were below the age-specific recommendations, as per the Sleep Duration Recommendations of the National Sleep Foundation [30]. Nevertheless, there was not much difference when compared to the results of a nationwide survey among Japanese children [3]. Hence, the nocturnal sleep duration of the children in the present study can be interpreted as generalizable for Japan. This problem was also noted in a report submitted to the CRC [31]. Furthermore, as mentioned in the introduction, this is a problem that the CRC recommends be addressed [5].

In this study, after confirming the validity of each latent variable by CFA (Figure 2), it was confirmed that the hypothetical model of ‘NSC’ → ‘parental lifestyle’ → ‘PA/ST’ → ‘SH’ should be adopted (Figure 3). Moreover, in the results of the simultaneous multiple-group SEM considering sex (Figure 4) and school stage (Figure 5), the CIM was adopted in both models. Thus, although these models were judged to have the same model structure regardless of sex and school stage, the estimated values cannot be said to be equal. Among them, it was the school stage difference of ‘ST’ → ‘SH’ that showed a relatively large difference in the estimated values (elementary school = 0.26, junior high school = 0.56). This may be because possession of smartphones is different between elementary and junior high school students in Japan. According to a survey conducted by the Tokyo Metropolitan Government on 2000 children, the smartphone ownership rate is 34.6% for upper elementary school students and 75.4% for junior high school students [32]. However, because of the sample size, notwithstanding a significant difference, the difference between most estimated values was minuscule. Moreover, it was confirmed that there were no significant differences in the estimated values of ‘PA’ → ’SH’ depending on sex and school stage. These results indicated that children’s SH were affected by their PA and ST, as well as being influenced by their parents’ lifestyle and the NSC that surrounded them.

Wright et al. [33] reported that adults participating in a week-long camp with exposure to sunlight and natural light alone saw a phase advance of melatonin (a sleep-inducing hormone) rhythm. The same findings have been confirmed in studies that involved long-term camps (e.g., 30 nights and 31 days) for children [34,35] and in an educational program that involved moving residence to a mountain village [36]. Needless to say, light stimulation during daytime [37], a dark environment at night [38,39,40], and moderate PA [41,42], which is effective in adjusting the biological rhythm, are guaranteed during camps and mountain village educational programs. Additionally, these programs also limited opportunities for ST, reducing it to nothing. Therefore, the pathway of ‘PA/ST’ → ‘SH’ in this study was confirmed at a practical level.

The American Academy of Pediatrics [43] recommends that school start times should be no earlier than 8:30 a.m., but, according to a study by Thacher and Onyper [44], the desired effect cannot be expected to occur by itself. In their study, the start time of school was delayed by 45 min to ensure sufficient sleeping time and its effect on sleep, behavior, and achievement in high school was verified. According to the results, six months after the intervention, sleep duration increased by about 20 min. However, after one year, bedtime and sleep duration returned to the original level, and no improvement was observed in academic performance. These results suggest that it is difficult to achieve persistent effects by only relying on direct means of intervention. In other words, the results of our study suggest that it is necessary to work on the pathway of ‘parental lifestyle’ → ‘PA/ST’.

Regarding the ties of local communities in Japan, neighborhood relationships decreased from 52.8% in 1975 to 10.7% in 2007, and neighborhoods that cooperated daily were reported to be 65.7% [45]. These results suggest that NSC has changed drastically over the last few decades and that building social connections is an issue for Japanese society. At the same time, it can be said that the present concerns regarding children’s SH in Japan, where social connection is an issue, are inevitable.

In addition, the incidence of poverty among children has become a social problem in Japan. According to a report by the OECD [46], the relative poverty rate of Japanese children (17 years old and under) was 13.9% in 2015 and was the 10th highest among 34 developed countries. De Clercq et al. [47] showed the effects of cultural and social capital on dietary habits, and Pinxten and Lievens [48] confirmed that the aforementioned types of capital contributed to physical and mental health issues. Moreover, the concept of time poverty pointed out by Vickery [49] cannot be ignored. These reports specify the need to consider not only NSC but also economic, cultural, and time capital as factors that determine the lifestyle of children and their parents. Although the estimated value of ‘NSC’ → ‘parental lifestyle’ was significant, it was not necessarily high because the effect of capital other than NSC was not considered.

As described above, in the present study, NSC was found to influence children’s SH through parental lifestyle and children’s PA/ST. Therefore, for improving the SHs of children over many years, direct interventions and measures to improve parents’ lifestyles and the NSC surrounding them would be effective. We consider this an important study finding. This can be understood along the lines of Bronfenbrenner’s model, which states that the development of individual children is influenced by their parents, and the NSC around them is shaped by the social context determined by interactions over time [6]. This study differs from many previous studies on child development in that it obtained evidence for countermeasures by evaluating the relevant factors from multiple dimensions, and includes the influence of specific social factors in the analytical model.

## 5. Limitations

This study had at least three limitations, and we would like to direct the focus of future studies towards addressing them. First, all data in this study were collected using a self-reported questionnaire, as this is an effective way to collect data from large sample sizes. However, a more detailed examination based on objective data is required. Second, factors other than NSC, which affect the lifestyle of children and their parents, were not examined. In this regard, a model verification that accounts for economic, cultural, and time capital should provide future directions. Third, we were not able to examine the relationship between the SH of children and health problems. In Japan, after the high economic growth period, people have been worried about ‘physical disorders’ among children. This issue was also indicated in the report on citizens and NGOs regarding the Rights of the Child [31]. Likewise, the final considerations by the CRC on this topic recommended that measures be taken to ensure that children enjoy their childhood, without any harm to their development caused by the competitive nature of society [5]. Therefore, future study topics should not only include lifestyle habits, such as children’s SH, PA, and ST, but also the health issues associated with them.

## 6. Conclusions

This study examined the pathways linked to SH among children and adolescents, regardless of sex and/or school stage, enrolled in the third grade or above at all public elementary and public junior high schools in Setagaya-ku, Tokyo. In total, 22,385 pairs of child–parent responses were used. The results confirmed that the NSC surrounding parental pathways had links to children’s SH through parental lifestyle and children’s PA and/or ST. Thus, we concluded that it is necessary to provide direct interventions and take additional measures to protect parents’ lifestyle and their NSC, to solve persistent sleep problems in children.

## Figures and Tables

**Figure 1 ijerph-18-06309-f001:**
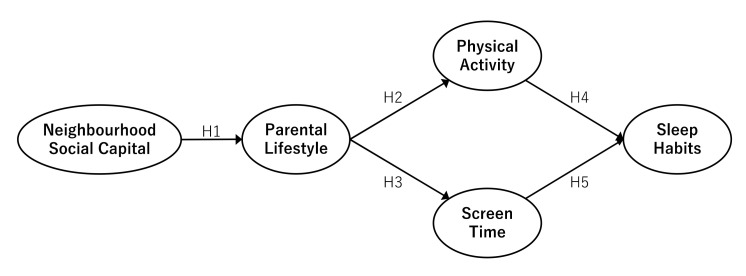
Hypothesized model.

**Figure 2 ijerph-18-06309-f002:**
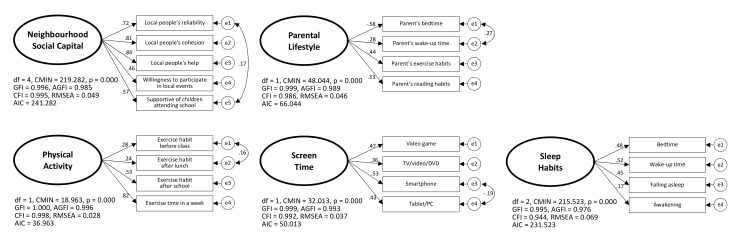
Confirmatory factor analysis of each latent variable.

**Figure 3 ijerph-18-06309-f003:**
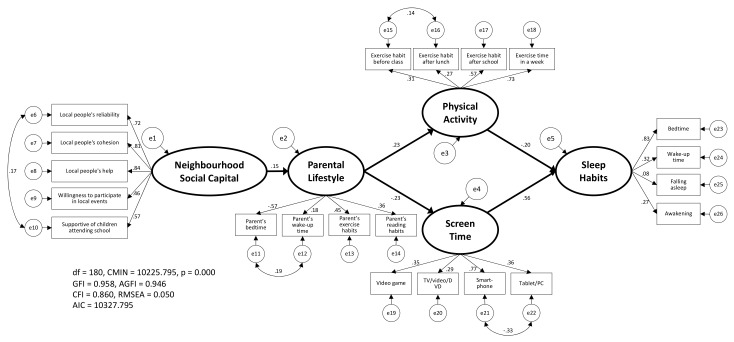
The pathways linking to children’s sleep habits by all samples. This model was verified by structural equation modeling using the bootstrap method. All estimated values in this figure are significant.

**Figure 4 ijerph-18-06309-f004:**
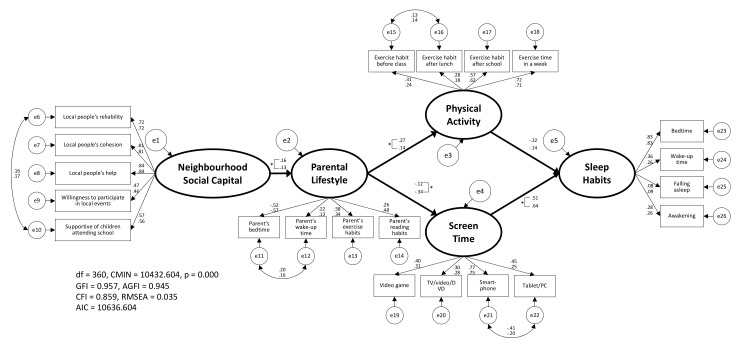
The pathways linking children’s sleep habits by sex. This model was verified by structural equation modeling using the bootstrap method. The numbers in Figure 4 pertain to upper: boys, and lower: girls, and all estimated values are significant. * The difference between boys and girls is significant at *p* < 0.05.

**Figure 5 ijerph-18-06309-f005:**
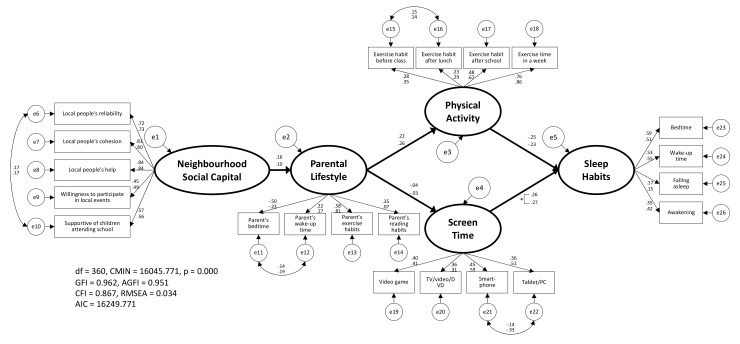
The pathways linking children’s sleep habits by school stage. This model was verified by structural equation modeling using the bootstrap method. The numbers in Figure 5 pertain to upper: elementary school and lower: junior high school, and all estimates, excluding the Parental Lifestyle → Screen Time values, are significant. * The difference between elementary school and junior high school is significant at *p* < 0.05.

**Table 1 ijerph-18-06309-t001:** Sleep habits, physical activity, and screen time of children by sex and grade level.

*n* = 22,385	Boys			Girls		
	ES 3rd–4th	ES 5th–6th	JHS 7th–9th	ES 3rd–4th	ES 5th–6th	JHS 7th–9th
	*n* = 4348	*n* = 3968	*n* = 3327	*n* = 4002	*n* = 3760	*n* = 2980
Sleep Habits						
Bedtime (mean ± SD, h:m)	21:47 ± 43	22:11 ± 48	23:28 ± 82	21:52 ± 43	22:19 ± 48	23:35 ± 76
Wake-up time (mean ± SD, h:m)	6:51 ± 33	6:56 ± 36	6:58 ± 42	6:54 ± 32	6:57 ± 38	6:52 ± 39
Sleep duration (mean ± SD, h^h^m^m^)	9^h^04^m^ ± 44^m^	8^h^44^m^ ± 49^m^	7^h^29^m^ ± 82^m^	9^h^01^m^ ± 45^m^	8^h^38^m^ ± 51^m^	7^h^17^m^ ± 75^m^
Sleep problems ^1^						
Sleep onset (%)	19.7	19.8	19.5	19.6	19.2	19.5
Bad awakening (%)	29.2	35.7	44.0	33.8	40.9	50.5
Nocturnal awakening (%)	12.5	11.4	9.1	11.5	9.2	8.2
Physical Activity ^1^						
Exercise habit before class (%)	40.5	31.9	16.7	23.6	18.4	13.2
Exercise habit after lunch (%)	87.0	78.6	55.5	73.4	57.2	30.3
Exercise habit after school (%)	72.6	66.9	68.2	66.9	62.8	55.3
Exercise time in a week (mean ± D, h^h^m^m^)	9^h^22^m^ ± 461^m^	8^h^34^m^ ± 450^m^	8^h^28^m^ ± 451^m^	5^h^49^m^ ± 327^m^	5^h^17^m^ ± 331^m^	6^h^22^m^ ± 417^m^
Screen Time						
Video game (mean ± D, h^h^m^m^)	0^h^39^m^ ± 60^m^	0^h^44^m^ ± 77^m^	1^h^16^m^ ± 118^m^	0^h^19^m^ ± 48^m^	0^h^21^m^ ± 46^m^	0^h^34^m^ ± 87^m^
TV/video/DVD (mean ± D, h^h^m^m^)	1^h^19^m^ ± 79^m^	1^h^24^m^ ± 88^m^	1^h^45^m^ ± 117^m^	1^h^28^m^ ± 100^m^	1^h^32^m^ ± 89^m^	1^h^59^m^ ± 117^m^
Smart phone (mean ± D, h^h^m^m^)	0^h^12^m^ ± 39^m^	0^h^27^m^ ± 64^m^	2^h^11^m^ ± 144^m^	0^h^13^m^ ± 39^m^	0^h^36^m^ ± 76^m^	2^h^30^m^ ± 139^m^
Tablet/PC (mean ± D, h^h^m^m^)	0h25^m^ ± 52^m^	0^h^34^m^ ± 61^m^	0^h^52^m^ ± 112^m^	0^h^20^m^ ± 45^m^	0^h^27^m^ ± 62^m^	0^h^34^m^ ± 83^m^
Screen time of 2 h or more (%) ^2^	59.6	70.8	91.1	52.9	65.0	89.5

Note: ES = elementary school, JHS = junior high school, ^1^ Percentage of those who answered ‘yes’; ^2^ Percentage of those who engage in screen time of 2 h or more.

**Table 2 ijerph-18-06309-t002:** Parental lifestyle and neighborhood social capital by sex and grade level of their child.

*n* = 22,385	Boy’s Parent			Girl’s Parent		
	ES 3rd–4th	ES 5th–6th	JHS 7th–9th	ES 3rd–4th	ES 5th–6th	JHS 7th–9th
	*n* = 4348	*n* = 3968	*n* = 3327	*n* = 4002	*n* = 3760	*n* = 2980
Lifestyle						
Bedtime (mean ± SD, h:m)	22:58 ± 80	23:09 ± 73	23:34 ± 69	23:00 ± 77	23:14 ± 72	23:39 ± 72
Wake-up time (mean ± SD, h:m)	6:25 ± 46	6:27 ± 46	6:26 ± 53	6:25 ± 44	6:26 ± 44	6:26 ± 54
Sleep duration (mean ± SD, h^h^m^m^)	7^h^26^m^ ± 85^m^	7^h^17^m^ ± 82^m^	6^h^52^m^ ± 73^m^	7^h^24^m^ ± 85^m^	7^h^11^m^ ± 79^m^	6^h^46^m^ ± 68^m^
Exercise habits						
1: Several days per month (%)	41.2	40.5	38.6	44.6	43.7	46.2
2: 1 or 2 days per week (%)	27.2	27.2	25.1	28.0	28.9	23.5
3: 3 or 4 days per week (%)	14.5	14.6	15.4	14.3	13.8	14.4
4: 5 or 6 days per week (%)	7.5	8.0	12.7	7.0	6.4	9.5
5: Every day (%)	9.6	9.7	8.2	6.2	7.2	6.5
Reading habits						
1: Did not (%)	27.9	30.3	41.5	26.5	27.3	41.7
2: 1 book per month (%)	21.9	22.8	23.8	19.3	21.5	22.3
3: 2–3 books per month (%)	27.6	27.3	23.0	27.7	28.1	23.6
4: 4–7 books per month (%)	13.4	12.2	7.5	14.5	13.3	8.1
5: 8–11 books per month (%)	4.0	3.7	1.7	5.1	4.2	2.0
6: More than 12 books per month (%)	5.2	3.8	2.5	6.8	5.7	2.3
Neighborhood Social Capital ^1^						
Local people are reliable.	65.3	64.9	64.6	64.5	64.1	64.5
Local people have a strong cohesion.	38.9	39.8	38.5	37.4	39.3	39.7
Local people are happy to help neighbors.	42.8	44.0	43.1	42.5	43.0	42.3
I want to participate in local events.	51.8	45.1	43.5	53.0	47.6	44.7
Local people are supportive of children attending school.	65.1	64.5	60.6	64.4	65.0	60.0

Note: ES = elementary school, JHS = junior high school, ^1^ Percentage of those who answered ‘yes’ and/or ‘somewhat yes’.

**Table 3 ijerph-18-06309-t003:** Goodness-of-fit indices for each structural equation model.

	df	CMIN	*p*	GFI	AGFI	CFI	RMSEA	AIC
*Sex*								
Configural invariance model	360	10,432.604	0.000	0.957	0.945	0.859	0.035	10,636.604
Equality constraints model	365	10,638.532	0.000	0.956	0.944	0.856	0.035	10,832.532
*School Stage*								
Configural invariance model	360	16,045.771	0.000	0.962	0.951	0.867	0.034	16,249.771
Equality constraints model	365	16,588.953	0.000	0.961	0.950	0.864	0.034	16,588.953

Note. GFI = goodness-of-fit index, AGFI = adjusted goodness-of-fit index, CFI = comparative fit index, RMSEA = root mean square error of approximation, AIC = Akaike information criterion, CMIN = chi-square, df = degrees of freedom.

## Data Availability

The data that supports the findings of this study are available from the corresponding author, upon reasonable request.

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
