# Peer review of "The Pathways Linking to Sleep Habits among Children and Adolescents: A Complete Survey at Setagaya-ku, Tokyo"

_ijerph, 2021, doi:10.3390/ijerph18126309_

Round 1
Reviewer 1 Report
I appreciate the thoughtful edits to the manuscript. My only suggestion is regarding the models. The values are placed on top of the lines making them hard to read. I suggest that these be moved slightly to one side for better interpretation.
Author Response
Thank you very much for your time. We are thankful for the time and energy you expended. Receiving your comment, we revised Figure 2, 3, 4, and 5. Additionally, we have checked again our manuscript spells by English native speaker.
Reviewer 2 Report
Nice work! I think the revision significantly improved the quality of the paper.
Author Response
Thank you very much for your time. We are thankful for the time and energy you expended. Receiving your comment, we have checked again our manuscript spells by English native speaker.
This manuscript is a resubmission of an earlier submission. The following is a list of the peer review reports and author responses from that submission.
Round 1
Reviewer 1 Report
Sleep is a process that contributes to all aspects of human development and interaction and research over the last two decades has made it evident that the quality and quantity of sleep persons around the world are getting has been reduced to the point that it is a public health concern. Research that aims to understand what contributes to sleep and sleep habits critical if we are to find public health messaging that works to improve sleep. The large sample size, the inclusion of children and parents, the comprehensive view of potential contributors to sleep, and the use of structure equation modeling are all strengths of the study. However, there are some significant concerns that must be addressed before this article should be published.
- There are many validated sleep assessments and so it is unclear as to why the researchers decided to create their own assessment. This need may have been due to translation issues but creating your own assessment can be a slippery slope and so when it is done it is important to be sure that details of the assessment are clear. There is simply not enough detail about the assessments provided to replicate the study or to have confidence in the results. Providing the questions or examples of the questions would be a good start. The scoring of the survey is included in the data analysis section, but that should appear in the method. I was unable to find any information about what questions made up the sleep problem score. These methodological issues should be addressed before the paper could potentially be published.
- The literature review does not provide a review of all the areas that are assessed in the study. There is not a review of why or how physical activity and screen time, two primary variables, are linked to sleep. More justification of the variables selected to include should be provided in the introduction. There is some of this information provided in the Hypothetical model section, but it should be included in the introduction in some degree as well.
- I am curious as to why an assessment of parents’ screen time was not included. Given that the hypothesized model assumes that parent lifestyle, such as reading habits, leads to the amount of screen time the child is exposed to, I would expect the parents’ screen time habits to be an important component to consider.
- Table 1 is not aligned properly, and it makes it difficult to interpret. For example, the last two columns do not have information showing up in the top line or in the bottom two or three lines. The note does not contain all of the abbreviations (ES; JHS). Although not as difficult to interpret as Table 1, Table 2 numbers should be on the same line as the statement that they associate with.
- I found the literature noted in the discussions to be helpful in understanding the issues of Japan. I believe that providing this information in the initial introduction would be beneficial.
- There is not information provided on what the sleep duration of South Koreans is, so it is unclear what the first sentence in the second paragraph of the introduction is trying to say. Does the research show that South Koreans are getting less sleep than all other countries? If so, that needs to be stated.
Using the bioecological theory to frame this work may help to tell the story that the authors are trying to relay.
Author Response
Thank you for your time. Please see the attachment.

Reviewer 2 Report
This paper investigates a very timely and important question. I think the paper reads quite well in general. I personally enjoyed reading it very much. The results section is well organized and the description of the method is clear. I appreciate the authors' efforts in conducting detailed analyses and controlling for gender and school stage. While I believe this paper could make contributions to the literature, I would give some suggestions for further improvement, especially for the introduction section.
The authors should provide clear definitions for each concepts (e.g., social capital). As this paper focuses mainly on the causal structure of “neighbourhood social capital (NSC)” -> “parental lifestyle” -> “child’s physical activity (PA)/screen time (ST)” -> “child’s sleep habits”, the authors should also present relevant studies on this structure. For example, why social capital will influence the parental lifestyle and consequently children's PA and ST....
For the method, the authors should justify why adding two items to the original scale for NSC.
Author Response

(The authors gave the same response as above.)

Round 2
Reviewer 1 Report
The authors responded to the suggested edits appropriately. The revised manuscript is stronger and the information provided allows for replication.